# Roles of Nitric Oxide in Brain Ischemia and Reperfusion

**DOI:** 10.3390/ijms23084243

**Published:** 2022-04-11

**Authors:** Yijie Wang, Fenfang Hong, Shulong Yang

**Affiliations:** 1Experimental Center of Pathogen Biology, College of Medicine, Nanchang University, Nanchang 330047, China; yijiewyj0406@163.com; 2Queen Mary University of London Nanchang Joint Program, Medical College, Nanchang University, Nanchang 330006, China; 3Department of Physiology, Fuzhou Medical College, Nanchang University, Fuzhou 344099, China

**Keywords:** brain ischemia and reperfusion, nitric oxide, ischemic stroke

## Abstract

Brain ischemia and reperfusion (I/R) is one of the most severe clinical manifestations of ischemic stroke, placing a significant burden on both individuals and society. The only FDA-approved clinical treatment for ischemic stroke is tissue plasminogen activator (t-PA), which rapidly restores cerebral blood flow but can have severe side effects. The complex pathological process of brain I/R has been well-established in the past few years, including energy metabolism disorders, cellular acidosis, doubling of the synthesis or release of excitotoxic amino acids, intracellular calcium homeostasis, free radical production, and activation of apoptotic genes. Recently, accumulating evidence has shown that NO may be strongly related to brain I/R and involved in complex pathological processes. This review focuses on the role of endogenous NO in pathological processes in brain I/R, including neuronal cell death and blood brain barrier disruption, to explore how NO impacts specific signaling cascades and contributes to brain I/R injury. Moreover, NO can rapidly react with superoxide to produce peroxynitrite, which may also mediate brain I/R injury, which is discussed here. Finally, we reveal several therapeutic approaches strongly associated with NO and discuss their potential as a clinical treatment for ischemic stroke.

## 1. Introduction

Ischemia and reperfusion (I/R) is a pathological process defined as a deficiency in the blood supply to an organ followed by restoration of perfusion and the oxygen supply [1]. Under clinical conditions, I/R in the brain can be triggered by the formation of a blood clot in the cerebral artery, resulting in the disturbance of metabolic processes. After rapid recanalization through thrombolysis or mechanical thrombectomy, perfusion to the ischemic area can be recovered successfully; however, inflammation and brain damage can be aggravated during reperfusion, which is known as ischemia-reperfusion injury [2,3]. Cerebral ischemia-reperfusion injury is associated with severe clinical manifestations, including ischemic stroke, encephalopathy, perinatal hypoxia-ischemia, and Alzheimer’s disease, in which ischemic stroke is one of the most prevalent diseases strongly associated with I/R [4,5]. As the major subtype of stroke, ischemic stroke is considered a disaster due to the high mortality and disability, which places a significant burden on both families and society [6]. However, effective treatments for ischemic stroke are still lacking, which may be caused by the complicated pathophysiological process of I/R. When blood flow in the cerebral arteries is restricted, ATP synthesis is significantly inhibited by the lack of glucose and oxygen supply, resulting in lactate accumulation and ion pump dysfunction. After, membrane depolarization is triggered by an imbalanced ion concentration, inducing the release of large amounts of excitotoxic amino acids, including glutamate [7]. Through binding to a specific receptor, glutamate activates ion channels and induces Ca^2+^ influx, which causes the disturbance of downstream signaling pathways and finally leads to disastrous outcomes, including neuronal cell death, inflammation, and microvascular dysfunction [8,9,10]. During reperfusion, despite restoration of the blood supply in the ischemic area, numerous free radicals, including RNS (reactive nitrogen species) and ROS (reactive oxygen species), are produced at this stage, which are considered to be important factors that contribute to cerebral ischemia-reperfusion injury [7]. However, the molecular mechanism regarding the formation of cerebral ischemia-reperfusion injury remains elusive (Figure 1).

Nitric oxide (NO), as a neutral hydrophobic signaling molecule, can participate in a wide range of physiological and pathological processes. The diffusion rate of NO is limited by the fast scavenging reaction with oxyhemoglobin. Therefore, the actions of NO mainly occur locally within tissue due to its limited half-life (1–10 s) and diffusion distance (50–1000μm) [11]. It is mainly generated through an enzymatic pathway, which converts L-arginine and oxygen to NO by nitric oxide synthases (NOS). NOS contains three different isoforms, including endothelial NOS (eNOS), neuronal NOS (nNOS), and inducible NOS (iNOS), which are activated under different conditions and provide NO with different properties. eNOS and nNOS are calcium dependent, whereas iNOS is calcium independent. Under physiological conditions, eNOS is mainly activated to generate NO at a low level (less that 10 nmol/L) to maintain essential functions [12,13]. In contrast, nNOS and iNOS, activated by numerous stimulants, produce high concentrations of NO, causing tissue damage and dysfunction. In brain I/R, the increase in NO production is detected in both the ischemia and reperfusion stages. At the early stage of cerebral ischemia, due to transient cerebral hypoperfusion, eNOS is upregulated to produce small amounts of NO, which mediates vasodilation and protects the brain vasculature [14]. With the decline in the blood and oxygen supply, glutamate accumulates rapidly and activates the calcium channel, inducing the activation of calcium-dependent nNOS to produce large amounts of NO. During reperfusion, the expression of iNOS is upregulated, resulting in excessive NO accumulation and iNOS activity lasting for a long period in the brain. In brain I/R, the excessive NO produced by nNOS and iNOS is neurotoxic, which contributes to inflammation, cell death, and BBB damage, finally resulting in brain ischemia-reperfusion injury [13]. Compared with endogenous NO, exogenous NO produced by various NO donors or NO inhalation ameliorates brain ischemia-reperfusion injury by protecting the brain from neurotoxic NO, which may have potential as a therapeutic approach for brain I/R in the future [15,16,17].

In this review, we focus on the effects of endogenous NO in the pathological process of brain I/R, especially the signaling cascades modified by NO, which affect cell survival and BBB integrity. Except for NO, we are also interested in the role of peroxynitrite (ONOO^-^), produced by the rapid reaction of NO and superoxide in brain I/R. Moreover, due to the close relation between NO and brain I/R, NO represents a significant breakthrough in clinical therapy for brain I/R. Therefore, several potential therapeutic approaches to brain I/R that are related to NO are discussed in the present review.

## 2. Roles of NO in Brain I/R

### 2.1. Neuronal Cell Death

The NR2B-PSD95-nNOS signaling complex is the best characterized and most widely accepted pathway related to neural cell death in brain ischemia and reperfusion. The formation of this complex is initiated during the early stage of brain I/R. It has been acknowledged that the first discovered pathological process of brain ischemia is excitotoxicity, in which excessive glutamate is released rapidly and is not recycled [18,19]. Overactivated by accumulative glutamate, the N-methyl-D-aspartate receptor (NMDAR), a type of cation-permeable inotropic receptor, mediates the Ca2+ influx, leading to an abnormal increase in the calcium concentration in the cytosol. Depending on the calcium concentration, nNOS is translocated from the cytosol and forms a complex with NR2B, a subunit of NMDAR, through the scaffolding protein PSD95 (postsynaptic density protein 95), which brings nNOS close to the NMDAR channel pore to activate nNOS. Ultimately, a large amount of neurotoxic NO is produced by the stimulated nNOS and disturbs specific signaling pathways, resulting in cell death and I/R injury in the brain [20,21,22,23].

Excessive NO produced by nNOS modulates the MAPK (mitogen-activated protein kinases) signaling pathway through S-nitrosylation to induce cell apoptosis during brain I/R. In the rat brain I/R, mixed linage kinase 3 (MLK3), a member of serine/threonine MAPK kinase kinases (MAPKKKs), is modified by NO through S-nitrosylation during the early stage of reperfusion, which is abolished by 7-nitroindazole (7-NI), an inhibitor of nNOS, but not 2-amino-5,6-dihydro-6-methyl-4H-1,3-thiazine (AMT), an inhibitor of iNOS. Subsequently, the activity of MLK3 downstream signaling molecules, including MKK4/7 (mitogen-activated protein kinase kinase 4/7), JNK3 (c-jun N-terminal kinase 3), c-Jun, and Bcl-2, expression of Fas-L (Fas ligand), release of cytochrome c from mitochondria, and activation of caspase 3 is inhibited after administration of 7-NI [24]. Taken together, NO produced by nNOS modifies MLK3 through S-nitrosylation to facilitate its activation, which in turn promotes the downstream MKK4/7-JNK signaling pathway and then subsequently activates the intrinsic and extrinsic apoptosis pathways, finally resulting in cell apoptosis and brain damage. Similar to MLK3, apoptosis signal-regulating kinase 1 (ASK1) is a general mediator of cell death and a member of the MAPK family, which also regulates the MKK4/7-JNK signaling cascade [25]. Research has shown that ASK1 is activated by NO through S-nitrosylation, which subsequently stimulates its downstream MKK4/7-JNK signaling pathway and then activates the nuclear apoptosis pathway to induce cell death during brain I/R [26]. The activation of the above signaling molecules induced by NO is affected by 7-NI, suggesting that ASK1 and its downstream signaling cascades are activated by NO in the same manner with MLK3 to promote cell apoptosis in rat brain I/R. However, activity of the non-nuclear apoptosis pathway has not been shown. Moreover, it is interesting to note that NO independently activates several downstream signaling molecules in the MAPK pathway through S-nitrosylation to activate the apoptosis pathway in rat brain I/R. For example, several studies have demonstrated that JNK3 and p38 MAPK are directly stimulated by endogenous NO, which then activates the downstream pathways and promote cell death, respectively [27,28]. 

Except for the MAPK signaling pathway, NO produced by nNOS in brain I/R targets other signaling cascades. Ca2+/calmodulin-dependent protein kinase II (CaMKII) is abundant in the brain with broad substrates [29]. Activated by autophosphorylation, CaMKII regulates a wide range of physiological functions involving the synthesis of neurotransmitters and membrane current [30]. Yu et al. found that phosphorylation of CaMKII decreased significantly during reperfusion in rat brain I/R, accompanied by an increasing level of S-nitrosylation. Combined with the increasing activity of nNOS they observed, they demonstrated that in brain I/R, NO produced by nNOS inhibited the autophosphorylation of CaMKII through S-nitrosylation, which caused the dysfunction of CaMKII and led to brain damage [16]. However, previous studies have shown the opposite result, showing that CaMKII is activated by Ca2+ and CaM and subsequently phosphorylates nNOS to inhibit NO production during brain ischemia in rats [31]. Yu et al. demonstrated the distinct situations with the spatial-temporal hypothesis. In their opinion, nNOS is brought closer to the channel pore through the formation of a complex with NMDAR and PSD95; therefore, inactivation of CaMKII by nNOS precedes the phosphorylation of nNOS [16]. In addition, the different stages in brain I/R may also affect the interaction between nNOS and CaMKII.

Apart from modulating the signaling cascades associated with cell death, NO produced by nNOS may also play a role in the regulation of the NMDAR-PSD95-nNOS signaling pathway through positive feedback. Evidence shows that the activation of c-src is triggered by NO through S-nitrosylation during brain I/R, which subsequently phosphorylates a subunit of NMDAR to enhance its activation, leading to delayed neural injury [32] (Figure 2).

### 2.2. Blood Brain Barrier Disruption

Except for neuronal cell death, disruption of the blood brain barrier (BBB) is another critical pathophysiological process that occurs in brain I/R. BBB, a specific microvasculature in the central nervous system (CNS), is composed of endothelial cells (ECs), pericytes, astrocyte end-feet, and extracellular matrix (ECM) [33]. As the key component of BBB, ECs lining the cerebral blood vessels are tightly linked by tight junctions (TJs) and undergo transcytosis at an extremely low rate, which is a unique property of ECs in BBB compared with ECs in other tissues [34]. Therefore, BBB tightly regulates the CNS homeostasis and prevents the entrance of potential toxins. During brain I/R, BBB disruption occurs in both the ischemia and reperfusion stages, which may promote brain edema and trigger hemorrhagic transformation [35,36,37,38]. Although the mechanism of BBB leakage caused by brain I/R still remains elusive, accumulating evidence shows that it correlates with alterations in TJ expression, ion channel activity, and several inflammatory processes induced by I/R [39,40,41].

Matrix metalloproteinases (MMPs) are a group of proteolytic enzymes with zinc-dependent catalytic site, including MMP-2 and MMP-9, which are mainly expressed in the brain and significantly associated with BBB disruption. Activation of MMPs is triggered by cerebral ischemia, resulting in degradation of TJs and ECM surrounding the cerebral blood vessels and neurons in BBB, which finally leads to hyperpermeability of ECs and BBB disruption. Recently, despite the exact molecular mechanism remaining unknown, mounting evidence has confirmed that endogenous NO facilitates the BBB disruption induced by brain I/R. One of the most convincing hypotheses proposed by Gu Y et al. demonstrates that endogenous NO produced by NOS contributes to BBB disruption by modulating the activity of MMPs during brain I/R. They found that the permeability of BBB in rats increased significantly during cerebral ischemia and reperfusion, accompanied by the upregulation of MMPs and downregulation of tight junction protein zonula occludens (ZO)-1 in the rat brain, which could be reversed by L-NAME [42]. Most importantly, the expression of caveolin-1 (cav-1) was decreased in rats that underwent cerebral ischemia and reperfusion, which was reserved by treatment with L-NAME. As an integral membrane protein, cav-1 inhibits NO production through binding to NOS [43]. After knocking out cav-1, the permeability of BBB was higher than wild-type rats and the activity of MMPs was slightly reduced, which further confirmed the above results. Taken together, their research suggests that NO produced by NOS inhibits the expression of cav-1 to initiate the activation of MMPs, resulting in impairment of TJs and BBB disruption.

However, in the pathophysiological process of brain I/R, controversy regarding the effect of endogenous NO on BBB integrity still exists. Robertson SJ et al. found that P-glycoprotein (P-gp) was upregulated in immortalized rat brain endothelial cells (GPNTs) following hypoxia-reoxygenation, which was abolished by an eNOS inhibitor [44]. P-gp is a multidrug efflux transporter located in the endothelial cells lining the brain blood vessels that restricts the access of molecules across BBB. Therefore, it is suggested that eNOS facilitates the expression of P-gp by producing NO, which finally protects BBB against cerebral ischemia-reperfusion injury.

## 3. The Roles of Peroxynitrite in Brain I/R

Apart from NO, peroxynitrite is another free radical produced during brain I/R. Peroxynitrite is generated through rapid reaction of NO and superoxide anion (O_2_^−^) at the diffusion rate, and is more active than its precursors [45]. As a reactive oxidant, the half-life of peroxynitrite is extremely short and its generation is constrained by the consumption of NO and the decay of O_2_^−^. With its high penetrating capacity, peroxynitrite crosses the plasma membrane more easily than superoxide anion to oxidize intracellular molecules, resulting in protein tyrosine nitration, lipid peroxidation, mitochondria dysfunction, and DNA breakage [8]. Tyrosine nitration, one of the best understood intrinsic properties of peroxynitrite, refers to the addition of a nitro group (-NO) to the hydroxyl group of tyrosine residue accompanied by the generation of 3-nitrotyrosine (3-NT), the biomarker of peroxynitrite [46]. An increase in 3-NT has been detected in both brain ischemia-reperfusion models and ischemic stroke patients [47,48]. During brain I/R, peroxynitrite exerts strong cytotoxic effects by mediating neuronal cell death, inflammation, BBB disruption, hemorrhagic transformation, and so forth [13,49,50]. Recent research has shown that after treatment with peroxynitrite decomposition catalyst (PDC), BBB damage, hemorrhagic transformation, cerebral infarct, and neurological dysfunction are attenuated in rodent brain I/R models, indicating that peroxynitrite plays a crucial role in brain I/R and could represent a vital target in ischemic stroke [50].

### 3.1. Mitophagy Activation

Autophagy is defined as the process of “self-eating” to remove accumulated proteins or dysfunctional organelles [51,52]. As selective autophagy, mitophagy refers to the elimination of injured mitochondria, which is essential for maintaining mitochondrial homeostasis and regulating cell survival [53,54]. When mitochondria are damaged, their membrane permeability is altered first. After the formation of mitophagosomes wrapped by isolated membranes, fusion with lysosomes occurs due to the participation of specific proteins, including light chain 3 (LC3) and p62, resulting in the degradation of damaged mitochondria. In this process, Parkin, an important mediator of mitochondrial fission, is activated by PINK (PTEN-induced kinase) to facilitate its translocation from the cytosol to mitochondria, where it binds to p62 and ultimately leads to mitophagy [55]. Except for the Parkin-dependent mitophagy pathway, there are several Parkin-independent mitophagy pathways, including the BNIP3/NIX signaling pathway and FUNDC1-mediated mitophagy pathway. Without the translocation of Parkin, these proteins (BNIPS/NIX, FUNDC1), located on the outer mitochondrial membrane, bind to LC3 directly to facilitate mitophagy [11,56,57].

Recently, numerous studies have proven the strong relation between mitophagy and brain I/R, and shown that dual effects of mitophagy occur in brain I/R. It is reported that mitophagy activation can attenuate neuronal damage and inflammation induced by brain I/R [58,59]. However, the suppression of mitophagy activation via specific treatments also alleviates brain I/R injury [48,60]. The key to interpreting this paradox is the reperfusion stage, which acts as a crucial point, shifting the role of autophagy/mitophagy from protection to destruction [61]. During the reperfusion stage, mitophagy is activated through several signaling pathways and contributes to brain I/R injury [55,62]. Among these signaling cascades, PINK/Parkin-mediated mitophagy is well-established in brain I/R. Interestingly, research has shown that peroxynitrite initiates the PINK/Parkin signaling pathway and participates in mitophagy during the reperfusion stage. Evidence obtained from this study suggests that peroxynitrite promotes the recruitment of Drp1 (dynamin-related protein 1), a key factor of mitochondrial fragmentation, to facilitate the translocation of Parkin, resulting in the initiation of PINK/Parkin-mediated mitophagy (Figure 3A). After treating PDC during the reperfusion stage, inhibition of Drp1 and Parkin translocation was observed accompanied by attenuated cerebral infarcts and neuronal damage [48]. Furthermore, research has shown that rehmapicroside protects against brain I/R injury by scavenging peroxynitrite to inhibit peroxynitrite-mediated mitophagy [60]. Taken together, these results convey an important message that peroxynitrite can mediate brain I/R-induced mitophagy and could represent a potential target for ischemic stroke.

### 3.2. Hemorrhagic Transformation

Until now, tissue plasminogen activator (t-PA) has remained the only FDA-approved thrombolytic drug for ischemic stroke, with a limited therapeutic time window of 4.5 h [63]. Delayed t-PA treatment beyond 4.5 h can increase the risk of hemorrhagic transformation (HT), in which BBB disruption mediated by MMPs is an important process [40]. Recently, numerous studies have indicated that peroxynitrite may promote MMPs activation to mediate the delayed t-PA-induced HT. Evidence shows that after delayed t-PA treatment, the production of peroxynitrite is significantly increased accompanied by MMP-9/MMP-2 expression, which is reversed by PDC [50]. These results suggest that delayed t-PA treatment induces the synthesis of peroxynitrite to subsequently upregulate MMPs expression, resulting in BBB damage and HT. Baicalin, which directly targets peroxynitrite, was shown to attenuate HT by inhibiting peroxynitrite-mediated MMPs activation in an ischemic stroke model with delayed t-PA treatment, which further confirms the effect of peroxynitrite in brain I/R-induced HT [64]. Moreover, high mobility group box 1 protein (HMGB1), as an activator of the upregulation of MMP-9 expression, may be involved in peroxynitrite-mediated HT. In ischemic stroke with delayed t-PA treatment, the level of HMGB1 is significantly increased with 3-NT, which can be abolished by PDC [65]. The suppression of HMGB1 by HMGB1-binding heptamer peptide can significantly ameliorate HT [66]. In order to further investigate the correlation between HMGB1 and peroxynitrte, 3-morpholino-sydnonimine (SIN-1), a peroxynitrite donor, was injected in native rat brain, which showed the increased expression of HMGB1 and TLR2 (toll-liike receptor 2) [65]. Taken together, these results convey a novel concept that HT induced by delayed t-PA is mediated by the peroxynitrite/HMGB1/MMPs signaling pathway in ischemic stroke (Figure 3B).

**Figure 3 ijms-23-04243-f003:**
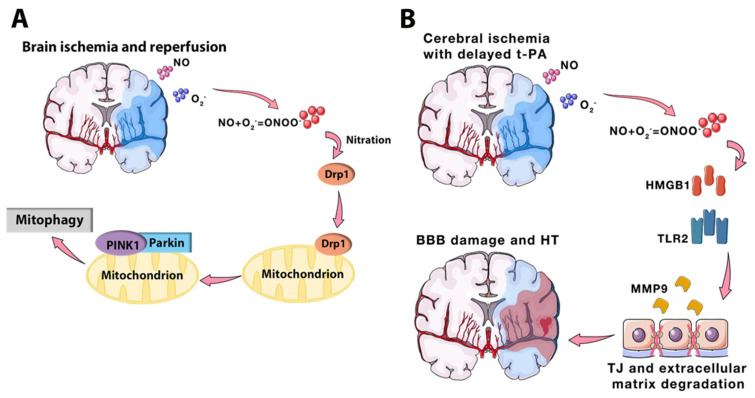
Effects of peroxynitrite in brain I/R. (**A**) In brain I/R, peroxynitrite produced during the reperfusion stage facilitates the recruitment of Drp1 via nitration to initiate PINK1/Parkin-mediated mitophagy, which may contribute to brain I/R injury [48]. (**B**) Another model of brain I/R is cerebral ischemia with delayed t-PA treatment, which facilitates the production of peroxynitrite. HMGB1 is activated by peroxynitrite and then upregulates the expression of MMP9 via TLR2, resulting in BBB disruption and HT [64,65]. HMGB1, high mobility group box 1 protein; TLR2, toll-like receptor 2; MMP 9, matrix metalloproteinase 9; TJ, tight junction; HT, hemorrhagic transformation; Drp1, dynamin-related protein 1; PINK, PTEN-induced kinase; BBB, blood brain barrier. Reproduced with permission from *Translational Stroke Research*, **2020**, *11*, 967–982. Copyright © 2022, Springer Science Bussiness Media, LLC, part of Springer Nature.

## 4. Potential Therapeutic Approach Related to NO

### 4.1. Ischemic Preconditioning

Ischemic preconditioning (IPC), first discovered in myocardium, refers to the sublethal ischemia induced by repetitive cycles of brief ischemia-reperfusion following severe ischemia in the same organ [67]. It is suggested that IPC induces ischemic tolerance through activation of the endogenous protective mechanism to protect against subsequent ischemia [68]. Several studies have shown that NO may mediate IPC neuroprotection. Mounting evidence shows that IPC can reduce the cerebral infarct size and neuronal cell death in the brain I/R model. Wang M et al. found that IPC upregulated the phosphorylation of CaMKII and nNOS, and downregulated the expression of c-jun and FasL, suggesting that IPC activates CaMKII to subsequently upregulate the phosphorylation of nNOS to reduce nNOS activity, finally leading to downregulation of the expression of c-jun and FasL to alleviate neuronal cell death [68]. Moreover, DDAH-1 (dimethylarginine dimethylamino hydrolase-1), capable of eliminating AMDA (asymmetric dimethylarginine) to facilitate NO synthesis, may play a critical role in IPC neuroprotection. Evidence obtained from a DDAH-1 knock-out rat model showed that brain damage could not be attenuated effectively after pretreatment with IPC compared with wild-type rats, indicating that DDAH-1 may mediate IPC neuroprotection. HIF-1α (hypoxia-inducible factor-1α), the downstream molecule of DDAH-1, is upregulated after pretreatment with IPC. However, after IPC pretreatment, the expression of HIF-1α and its target genes is affected in DDAH-1 knock-out rats. Taken together, this indicates that IPC upregulates the expression of DDAH-1 to activate HIF-1α and its target genes to regulate NO synthesis, resulting in ischemic tolerance [69].

### 4.2. Ischemic Postconditioning

Ischemic postconditioning is defined as a short period of sublethal ischemia induced in an organ to protect against prior ischemia that occurred in the same organ [70]. In brain I/R, ischemic postconditioning is conducted by a series of repetitive cycles composed of brief reperfusion and occlusion of bilateral common carotid arteries during the reperfusion stage [71]. Accumulating studies have confirmed that ischemic postconditioning can protect against brain injury in rat models of cerebral I/R. These studies suggest that ischemic postconditioning can reduce the cerebral infarct size, improve neurological deficits, decrease delayed neuronal death, ameliorate inflammation in which NO may be involved, and mediate ischemic postconditioning neuroprotection. Previous research showed that ischemic postconditioning significantly facilitated NO synthesis and improved the CBF in a global cerebral I/R rat model. The concentration of NO increased slowly during the early period and then reached a peak within 2 h, which is almost consistent with the trend of CBF. After pretreatment with 7-NI, a selective inhibitor of nNOS, NO production was inhibited accompanied by abolishment of ischemic postconditioning neuroprotection, suggesting that nNOS may be involved in ischemic postconditioning to protect against brain I/R injury [71]. In addition, a novel intervention called remote ischemic postconditioning (RIPoC), extending the definition of ischemic postconditioning, refers to the ischemic postconditioning induced in a distant organ to protect against ischemia occurring in an important organ. Previous studies have confirmed that RIPoC can reduce the cerebral infarct size, regulate the expression of apoptotic genes, decrease delayed neuronal cell death, and improve the spatial learning ability in cerebral I/R rat models [72,73]. Evidence shows that RIPoC upregulates the expression of Akt and eNOS, and the protective effect of RIPoC is abolished by non-selective NOS inhibitor or PI3K inhibitor, suggesting that RIPoC may activate the PI3K/Akt signaling pathway to upregulate the expression of eNOS to finally protect against brain I/R injury [73]. However, Leger PL et al. showed that ischemic postconditioning failed to protect against brain I/R injury in a neonatal cerebral stroke model. Ischemic postconditioning cannot reduce the cerebral infarct size in neonatal cerebral stroke, which may contribute to incomplete reperfusion and the absence of hyperemia [74].

### 4.3. Inhalation of NO

NO, as a potent vasodilator, can regulate the blood flow under physiological condition. Exogenous NO, delivered by inhalation, plays an important role in ameliorating I/R injury in many organs, including the brain [75]. Due to its specific characteristics, inhaled NO has been extensively investigated in pulmonary diseases, including COVID-19, and has been approved to treat pulmonary diseases associated with pulmonary vasoconstriction, including acute respiratory distress, chronic obstructive pulmonary disease, and primary pulmonary hypertension in newborns [76,77,78]. Recently, several studies have focused on the effect of inhaled NO on ischemic stroke and shown its potential to protect the brain against I/R injury. Terpolilli et al. demonstrated that inhaled NO, administered at the onset of the ischemia stage, not only reduced the infarct volume and improved the neurological function but also selectively dilated both venules and arterioles in ischemic penumbra [17]. In cerebral ischemia, inhaled NO significantly increased the penumbral CBF with a decreasing penumbral volume. Therefore, it is predicted that inhaled NO represents a promising therapeutic approach for ischemic stroke by improving collateral circulation and selectively increasing the blood supply in ischemic tissue. Moreover, different concentrations and durations of inhaled NO result in different efficacy, which has mainly been exhibited in a neonatal stroke model [79,80]. Evidence obtained from this study showed that inhaled NO (20 mmp) during the ischemia stage was strongly related to increased blood flow velocity and reduced cell damage. In contrast, inhaled NO administered during the reperfusion stage was detrimental, which may be attributed to immature collateral recruitment in neonates [80].

### 4.4. Strategies for Increasing eNOS Activity

Under physiological conditions, eNOS continuously produces low levels of NO to exert protective effects, including vasodilation, anti-platelet aggregation, and anti-inflammation. Based on this intrinsic property, increasing eNOS activity has become an important therapeutic approach for brain I/R. eNOS is activated by phosphorylation at multiple sites, among which Ser1177 is regarded as one of the most important sites that is phosphorylated by Akt [81]. Therefore, numerous therapeutic approaches for cerebral I/R injury are developed through activation of the PI3K/Akt/eNOS signaling pathway. Research has shown that Vitexin is able to increase the phosphorylation of Akt and eNOS to maintain BBB integrity in an ischemic stroke model, which can be abolished by PI3K inhibitor [82]. CXC195, a novel derivative of tetramethylpyrazine, can alleviate cerebral I/R injury in the same way as Vitexin [81]. Except for drugs, photostimulation with low-level light-emitting diode therapy (LED-T), which significantly improves the outcome of brain I/R, is strongly related to the activity of eNOS. Evidence shows that pretreatment with LED-T before brain I/R in mice improves the neurological function and reduces the infarct volume, which is mainly mediated by the phosphorylation of eNOS through activation of the PI3K/Akt pathway [83].

### 4.5. Strategy for Disrupting the NR2B-PSD95-nNOS Complex

Based on the information presented in Section 2.1, the high levels of NO produced by nNOS via excessive stimulation of NMDARs play a crucial role in cerebral I/R injury. Therefore, numerous drugs have been developed to inhibit the signaling complex we mentioned before. However, the inhibitors inhibiting NMDARs or nNOS also directly block their physiological functions, resulting in severe side effects. In order to reduce excessive NO production without affecting the normal function of NMDARs and nNOS, researchers designed a lentiviral vector expressing nNOS with a mutated N-terminal residue, which disrupted the interaction between nNOS and PSD95, resulting in significantly improved outcomes after brain I/R without severe side effects in mice [21]. After, a drug design study showed that ZL006, a small nonpeptidic molecule, blocks this complex by competing with PSD95 and selectively binding nNOS [21]. Recently, increasingly more drugs have been developed for cerebral I/R injury based on this theory, which effectively reduces cerebral I/R injury and solves the neurological side effects. For example, a study has shown that honokiol, a major bioactive component of *Magnolia officinalis*, is able to protect against ischemic stroke in mice. Honokiol not only decreases the interaction between nNOS and PSD95 but also inhibits the translocation of nNOS from the cytosol to the membrane, which finally disrupts NR2B-PSD95-nNOS complex and improves the outcome of ischemic stroke in mice [84].

## 5. Conclusions

In conclusion, we revealed the role of NO and peroxynitrite in the pathological process of brain I/R and several potential therapeutic approaches for brain I/R related to NO. During brain I/R, excessive NO, produced by abnormal stimulation of NOS, affects several signaling cascades associated with cell apoptosis and BBB damage, finally resulting in brain I/R injury. Moreover, high levels of NO participate in the formation of peroxynitrite, which further damages the brain via mitophagy activation and hemorrhagic transformation. Due to the intrinsic properties of NO, several potential therapeutic approaches were discussed in this review, including ischemic preconditioning, ischemic postconditioning, NO inhalation, strategies related to increasing eNOS activity, and disruption of the NR2B-PSD95-nNOS complex. Although these approaches have neuroprotective effects in cerebral I/R models, they still lack direct evidence and the exact mechanism required for approval as a treatment in clinical trials. Due to the significant role of NO in brain I/R, NO and its signaling pathway represent a novel target of ischemic stroke. The exact relationship between NO and brain I/R should be further explored in the future.

## Figures and Tables

**Figure 1 ijms-23-04243-f001:**
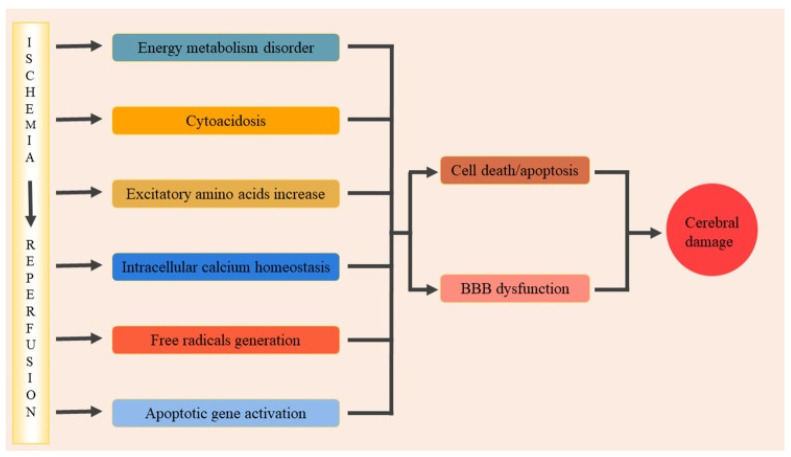
The main pathological processes of brain ischemia and reperfusion [7]. BBB, blood brain barrier. Coped with permission from Front. Mol. Neurosci. 2020, 13, 28. Copyright © 2022 Wu, Xiong, Wu, Ye, Jian, Zhi and Gu.

**Figure 2 ijms-23-04243-f002:**
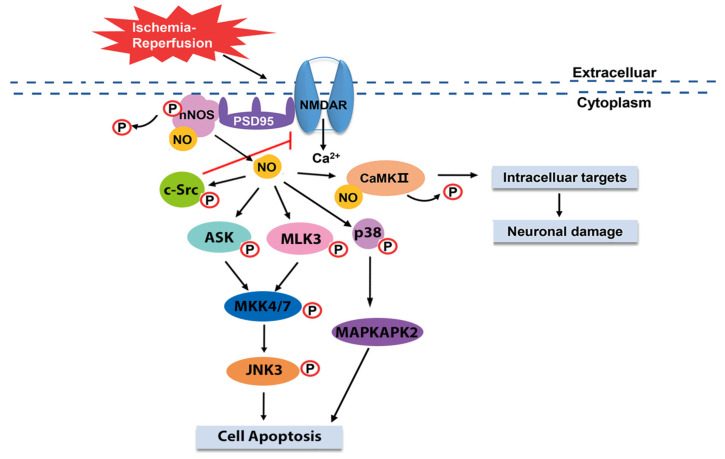
The NO signaling pathway, resulting in neuronal cell death during brain ischemia and reperfusion. During brain I/R, NMDAR forms a complex with nNOS through PSD95, resulting in nNOS activation and NO synthesis [20,21,22,23]. NO produced by nNOS activates c-Src through phosphorylation to inhibit the activity of NMDAR [32]. NO also phosphorylates downstream signaling molecules, including ASK, MLK3, and p38, which subsequently activates their downstream signaling proteins and finally stimulates the cell apoptosis pathway, resulting in cell death [24,25,26,27,28]. NO activates CaMKII through S-nitrosylation, resulting in neuronal damage [16]. NMDAR, N-methyl-D-aspartate receptor; nNOS, neuronal NOS; PSD95, postsynaptic density protein 95; ASK, apoptosis signal-regulating kinase; MLK3, mixed linage kinase 3; CaMKII, Ca2+/calmodulin-dependent protein kinase II; MKK4/7, mitogen-activated protein kinase kinase 4/7; JNK3, c-jun N-terminal kinase 3. Reproduced with permission from the *European Review for Medical and Pharmacological Sciences*, **2019**, *23*, 7674–7683. Copyright © 2022 Eur Rev Med Phamarcol Sci.

## Data Availability

Not applicable.

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
