# Peer review of "Roles of Nitric Oxide in Brain Ischemia and Reperfusion"

_ijms, 2022, doi:10.3390/ijms23084243_

Round 1
Reviewer 1 Report
This review entitled “Roles of nitric oxide in brain ischemia and reperfusion” by Wang et al., is an interesting clear and comprehensive review article that give an nice overview of nitric oxide involvement in brain ischemia and reperfusion. The review starts with the general concept for nitric oxide production and in parallel the brain injury associated with neuronal death and the blood brain barier disruption. Authors presented the roles of NO in neuronal cell death with a nice description of some signaling pathways involving nitrosylation regulation of MAPkinases and CaMKII. They discuss that blood brain barrier integrity is sensitive to NO via regulation of matrix metalloproteases (controversial). Another reactive oxygen species is also involved, the peroxinitrite that will modify tyrosine and lipids. The section 3.1 concerning mitophagy is not enough introduced in the introduction. Mitophagy activation is developed in a section specifically concerning peroxinitrite but Mitophagy is not only linked to peroxinitrite it would be nice to develop this more.A last section is dealing with Potential therapeutic approach related to NO, with a section of ischemic preconditioning, one of ischemic postconditioning one for inhalation of NO. The two first parts are well explained but the last one concerning NO inhalation could be expanded and more detailed.
Figures are appropriate easy to understand even if figure one is note necessary.
Some writing sound sometimes strange and could be ameliorated (some exemples L84, L95, L139, L291, L307).
The section conclusion also needs some improvements to be more complete.
Reviewer 2 Report
The authors wrote a review in where they detail the roles of nitric oxide and peroxinitrite in brain ischemia and reperfusion. The review is general and very-well written. It is easy to follow and understand for a non-expert in the topic like the reviewer. Thus, I would like to suggest the approve of this manuscript after few minor revisions, that will help the non-specialized reader.
- Is it possible to avoid the formation of NO in brain? Or at less reduce its concentration? (using different types of scavengers?) This question appeared to me as the review describes different pathways to produce NO.
- Maybe could be interesting to describe a little bit the biological properties of both NO and peroxinitrite. To give the reader some values of which is their typical concentration, their lifetimes, basic chemical properties
- Maybe could be interesting to expand the biological pathways as you can generate peroxinitrite. In this review is widely described how to generate NO, but I feel that there is a gap in the peroxinitrite generation.
